

# Human activities determine vegetation water use in the
# middle and lower reaches of arid areas
*Siyu Lu[a,b], Guofeng Zhu[a,b,*], Rui Li[a,b], Yinying Jiao[a,b], Gaojia Meng[a,b], Dongdong Qiu[a,b],*
*Yuwei Liu[a,b], Lei Wang[a,b], Xinrui Lin[a,b], Yuanxiao Xu[a,b], Qinqin Wang[a,b], Longhu Chen[a,b]*
**Affiliations:**
*[1] College of Geography and Environmental Science, Northwest Normal University, Lanzhou*
*730070, Gansu, China*
*[2] Shiyang River Ecological Environment Observation Station, Northwest Normal University,*
*Lanzhou 730070, Gansu, China*
*[*]Correspondence to: Guofeng Zhu (zhugf@nwnu.edu.cn)*
**Abstract:** In the middle and lower reaches of inland river basins of arid regions,
human-intensive exploitation directly determines the distribution patterns of plants in
arid areas and further determines the patterns of water use and the water cycle in arid
regions. However, human activities on vegetation water utilization and the influence
of the water cycle process and mechanism are not clear. In this study, seven
observation systems were set up to collect samples in the mountainous, oasis and
desert areas of the Shiyang River Basin, an arid inland river in central Asia. In order
to quantitatively assess the contribution of different potential water sources to plants,
stable isotopes of various water bodies in different geomorphic units of the basin were
analyzed. The results showed that precipitation and soil water were the main sources





of forest trees in mountainous areas, and the farmland vegetation in the middle and
lower reaches of the oasis mainly absorbed soil water supplied by irrigation. The
desert area forms vegetation in the ecological water transport area, and vegetation
mainly absorbs soil water, lake water and groundwater formed by ecological water
transport. On the whole, the water use patterns of plants in mountainous areas are not
affected by human activities fundamentally, the oasis area is mainly affected by
irrigation activities, and the inland river terminal lake area is mainly affected by
ecological water transport. Human activities determine the water use patterns in the
middle and lower reaches of inland rivers in arid areas.
**Keywords:** Arid areas;Stable isotope; MixSIAR model; Plant water use
**1 Introduction**

Water availability is one of the most important factors for the growth of

individual plants in terrestrial ecosystems (Boyer et al., 1982). Plant survival and
activities, as well as ecosystem stability is closely related to water availability (Zhou
et al., 2019). In arid and semi-arid areas, water is the main limiting factor for
vegetation development (Porporato et al., 2004). Due to water shortage, plant growth
is limited, but plants have strategies to prevent water loss and resist drought (Gupta et
al., 2020). Precipitation is one of the main sources of water (Zhao et al., 2018) and an
important climatic factor of vegetation change (Roca et al., 2004), which controls
plant community structure, composition and vegetation type (Weltzin et al., 2003).
The uneven distribution of precipitation leads to the extreme spatial and temporal
variability of soil moisture (Antunes et al., 2018). Under different precipitation



conditions, the water use strategies of vegetation will be different (Miller et al., 2001).
The distribution of precipitation and the depth of groundwater level control the spatial
pattern of soil moisture availability. This plays a crucial role in plant adaptation and
vegetation composition (Zhou et al., 2019). In addition, human beings have been
influencing the hydrological cycle since the beginning of civilization (Zhao et al.,
2020), and in recent years, human activities have changed the global and regional
environment and sustaining making influence (Yang et al., 2011). Human activities
are key factors affecting vegetation growth, which are manifested as affecting
vegetation types and vegetation degradation, etc. (Klein., 2012; Jiang et al., 2017).
Therefore, it is of great significance for ecological restoration and water resources
management to study the effects of human activities on vegetation water use patterns
under natural precipitation gradients in arid inland river basins.
Stable isotopes are natural traces widely distributed in natural water bodies and
have been widely used in plant water research (Ehleringer and Dawson., 1992). The
stable hydrogen and oxygen isotope characteristics of terrestrial ecosystems can
provide clear trace information for hydrological cycles in terrestrial ecosystems (Pan
et al.,2020). Generally, there is no stable isotope fractionation in the process of plant
water absorption. Thus, xylem water can reflect the isotopic composition of water
sources used by most plant species (Wershaw et al., 1966). Previous studies have
found that in arid areas, plants mainly absorb shallow soil water supplemented by
precipitation or deep groundwater supplemented by groundwater (Zeng et al., 2013),
and the water source used by individual plants will change over time (Nie et al., 2012).





Under water stress conditions, A steady, long-term source of water is essential for
plant survival. In addition, the utilization of rainwater by desert plants in arid and
semi-arid ecosystems is related to precipitation intensity. In areas with high annual
precipitation, growing artemisia ordosica and white thorn, have higher utilization
efficiency of shallow soil water, while in areas with low annual precipitation, they
mainly utilize deep soil water and groundwater (zhou et al., 2011). In addition, plant
water use behaviour can be linked to broader drought resistance strategies (Antunes et
al., 2018).
Although many studies have been conducted on plant water use in arid and
semi-arid environments, in the face of the strong impact of global change and human
activities, it is necessary to further clarify the change of vegetation water utilization in
mountainous areas, oases and deserts in arid areas and the impact of human activities
on vegetation water use patterns in arid areas. This study (1) determined the water
sources of different vegetation in mountainous areas, oases and deserts; (2) analyzed
the impact of human activities on vegetation water use patterns in arid areas; (3)
discussed the implications of vegetation water use strategies for water resources
management in arid areas.
**2 Materials and methods**
**2.1 Study area**
Shiyang River Basin (36°29′ - 39°27′ N, 101°41′ - 104°16′E) is located in the
arid region of northwest China, which is a typical temperate arid inland basin in China.
Shiyang River Basin is a temperate continental arid climate, which is controlled by



several atmospheric circulations of the westerly belt, eastern monsoon and plateau
monsoon (Zhang et al., 2008). The average annual temperature is 8.1℃, and the
average annual precipitation ranges from 82 to 692mm, 80% of which is concentrated
in summer. Annual evaporation ranges from 2000 to 2600mm (Wan et al., 2019).
Shiyang River is a typical inland water system with a total length of 250 kilometres.
From south to north, Shiyang River mainly includes Qilian mountain area in the south,
oasis area in the middle and a desert area in the north. Studies have found that the
water vapour transport track in this region is transported from the desert area in the
south to the mountainous area in the north through the oasis area (Zhu et al., 2019). In
addition, the annual precipitation in the three regions is 124-698mm, 83-124mm and
54-82mm from south to north, respectively (Ma et al., 2009), so the soil and
vegetation have obvious zonal characteristics (Wang et al., 2012). The main
vegetation in the mountainous area is Picea crassifolia, willow and ice grass,
Vegetation in the oasis area is mainly corn and other farmland vegetation and some
shrubs and the main vegetation in the desert area is a white thorn and Haloxylon
ammodendron.



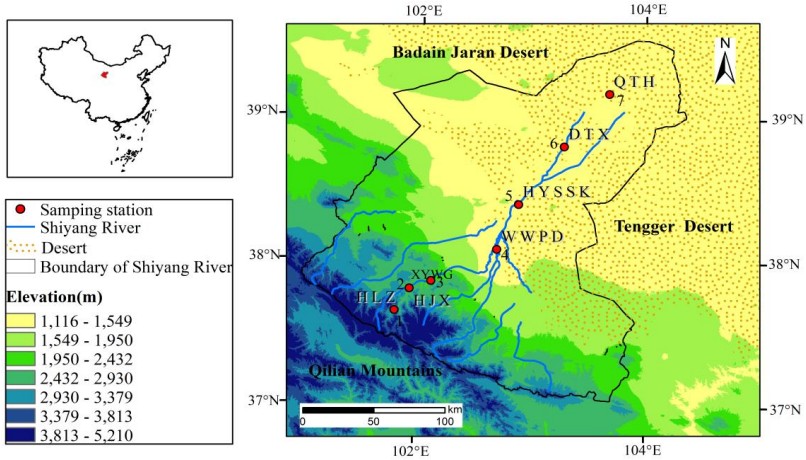

**Fig. 1.** Overview of the study area.

**2.2 Sample collection and measurement**

From 2017 to 2019, we collected samples of precipitation, soil, vegetation, and groundwater from seven stations in the Shiyang River Basin during the plant-growing season (April to November). Table 1 shows the summary data of the sample points. The selected sampling points are respectively distributed in the mountainous area, oasis area and desert area of Shiyang River Basin. There are three stations in the mountainous area (Hulinzhan(HLZ),Huajianxiang(HJX),Xiyingwugou(XYWG), three stations in the oasis area (Wuweipendi(WWPD), Hongyashanshuiku(HYSSK), Datanxiang(DTX)), and one station in the desert area (Qingtuhu(QTH)). Soil, vegetation and groundwater were sampled once a month. Rainwater samples were collected according to precipitation events by means of a rainwater gauge cylinder installed at the sampling point. Precipitation samples were collected immediately after the precipitation process. For continuous precipitation, we collect precipitation once a day. For plant collection, we selected stems more than 2 years old, took branches



about 0.35-0.5cm in diameter and 3-5cm in length, quickly stripped the epidermis and
phloem of the branches, retained the xylem, and immediately placed them into
sampling bottles for sealing. For groundwater collection, we collected groundwater
near the sampling point. In the vicinity of the plant sampling site, soil samples were
collected every 10cm of the soil using a soil drill, up to 100cm depth if conditions
permit. The collected soil samples were divided into two parts. The first part was
sealed in 10ml glass bottles with sealing film and stored at -18°C for subsequent
analysis of δD and δ$^{18}$O in the soil. The second part was placed in aluminium boxes
and dried in the laboratory to determine soil water content.
**Table 1.** Basic information about sampling points.

| Sample station | Longitude | Latitude | Elevation | Mean annual temperature (°C) | Annual precipitation (mm) |
|---|---|---|---|---|---|
| Hulinzhan(HLZ) | 101°50' | 37°41' | 2721 | 3.2 | 370 |
| Huajianxiang(HJX) | 102°00' | 37°50' | 2323 | 6.6 | 363.5 |
| Xiyingwugou(XYWG) | 102°11' | 37°53' | 2097 | 7.9 | 262.5 |
| Wuweipendi(WWPD) | 102°40' | 37°55' | 1531 | 10.2 | 186.5 |
| Hongyashanshuiku(HYSSK) | | | 1475 | | 113 |
| Datanxiang(DTX) | 103°13' | 38°47' | | | 113.2 |
| Qingtuhu(QTH) | 103°35' | 39°05' | 1300 | 7.8 | 110 |

**2.3 Isotopic composition and analysis of hydrogen and oxygen**



132  All samples were analyzed for δ2H and δ18O at the Stable Isotope Laboratory of

133 Northwest Normal University using a Liquid Water analyzer (DLT-100, Los Gatos

134 Research, USA). Soil water and vegetation water were extracted and analyzed by a

135 vacuum low-distillation device (LI-2100, LICA United Technology Limited, China).

136 The extraction accuracy of the low-temperature and low-pressure distillation device

137 was up to 98%. The measured values of isotopes are denoted by the symbol "δ" and

138 expressed as one-thousandth of the Vienna standard means sea water:

139   $\delta = [\frac{R_{sample}}{R_{standard}} - 1] \times 1000\%$       (1)

140  Where, $R_{sample}$ represents the ratio of $^{18}O/^{16}O$ or $D/^1H$ in the precipitation sample,

141 and $R_{standard}$ is the ratio of $^{18}O/^{16}O$ or $D/^1H$ in V-SMOW.

142 **2.4 Data analysis**

143  The MixSIAR isotope mixing model based on Bayesian theory was used to

144 identify soil water sources and quantitatively analyze the contribution ratio of

145 different water sources (Stock and Semmens., 2013). Bayesian mixed models have

146 advantages over simple linear mixed models in estimating the probability distribution

147 of source contributions (Zhu et al., 2021). In the MixSIAR model, the input of xylem

148 water and soil water isotope values in each soil layer were all original data, TDF data

149 was set as 0, and isotope fractionation did not occur by default. The operating length

150 of the Markov chain Monte Carlo (MCMC) was set as "extreme", and the error

151 structure was set as Rm. Soil water in different soil layers was considered the

152 potential water source for vegetation in arid areas. The classification of potential

153 water sources was based on the isotopic composition of soil water and soil water

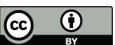



content. The shallow layer (0-20 cm) was greatly affected by precipitation, irrigation
water and evaporation, and the soil water content and isotopic composition of soil
water changed greatly. The changes in soil water content and soil water isotopic
composition in the middle layer (20-60 cm) were relatively small. The variation of
soil water content in the deep layer (60-100cm) was the least, and the isotopic
composition of soil water was stable. The $\delta 18O$ values of each potential water source
were brought into the MixSIAR model to determine the contribution ratio of each
potential water source.
**3 Results and analysis**
**3.1 Isotopic values of different water bodies**
**3.1.1 Precipitation isotope**

Precipitation gradually decreased from mountainous to desert areas, with

significant differences between $\delta D$ and $\delta^{18}O$ (Fig. 2). The annual precipitation of the
seven sampling sites was ranked as follows: HLZ > HJX > XYWG > WWPD >
HYSSK > DTX > QTH (Table 1). Because the Shiyang River Basin is located in the
inland region, it is difficult for warm and wet water vapor from the western Pacific
Ocean to reach it, and it is affected by secondary evaporation during the precipitation
process, which leads to the enrichment of precipitation isotopes in summer. In
September, the stable isotope values of precipitation begin to decrease. In the growing
season from April to November, the d-excess ranking of the seven sampling sites was
in the following order: HLZ (14.8‰) > HJX (12.5‰) > XYWG (12.4‰) > HYSSK
(8.8‰) > WWPD (8.7‰) > DTX (7.6‰) > QTH (5.7‰). The reason for these results



may be the intense evaporation of raindrops as they fall.

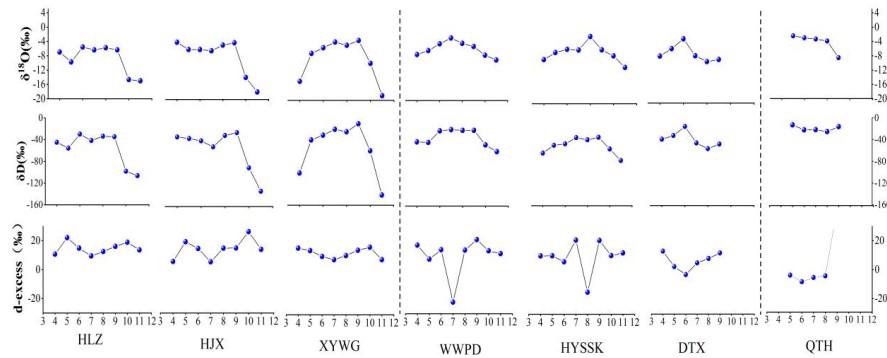


**Fig. 2.** Variation of δD(‰), δ18O(‰) and d-excess in vegetation growth season (April-November)
in mountainous, oasis and desert areas of the arid region. HLZ, HJX, XYWG, WWPD, HYSSK
and DTX, respectively, represent the Hulinzhan, Huajianxiang, Xiyingwugou, Wuweipendi,
Hongyashanshuiku, Datanxiang and Qingtuhu.

**3.1.2 Isotopic composition of soil water, groundwater and xylem water**

A linear relationship was established between δD and $\delta^{18}O$ in precipitation, soil
water, xylem water and groundwater samples at seven stations (Fig. 3). The slope and
intercept of LMWL at seven sampling points were all smaller than GMWL, because
Shiyang River Basin was located in the arid region of Northwest China, where
evaporation was intense. The slope ranking of LMWL of the seven sampling points
was QTH < DTX < HLZ < HYSSK < XYWG < WWPD< HLZ, indicating that the
evaporation in the desert area of Shiyang River Basin was the strongest, followed by
the oasis area, and mountain area was the weakest. The isotopic values of soil water in
mountainous areas (HLZ, HJX, XYWG) and oasis areas (WWPD, HYSSK) were
consistent with LMWL, indicating that precipitation in these areas may be the


potential water source for soil water recharge. However, the soil water isotopes in the
oasis area (DTX) and the desert area (QTH) were inconsistent with the LMWL,
indicating that the precipitation had less soil water recharge in these two areas.

With the increase of soil depth, the deviation of the soil water isotope from

LMWL gradually decreased. There were significant differences in the utilization of
soil water in different soil layers by vegetation in different locations with precipitation.
The $\delta^{18}O$ values of xylem water from the mountainous area to the oasis area to the
desert area (The $\delta^{18}O$ values of xyloxyme water from Qinghai spruce in HLZ, willow
in HJX, white poplar in XYWG, corn in WWPD, poplar in HYSSK, corn in DTX and
white prickly tree in QTH were -5.02‰, -4.64‰, -5.51‰, -7.60‰, -5.64‰, -5.45‰,
-1.59‰, respectively)are similar to the $\delta^{18}O$ values of soil water in the surface layer,
and the middle layer, respectively. The $\delta^{18}O$ values of xylem water were the highest in
the desert area (QTH) and the lowest in the oasis area (WWPD). These results
indicated that vegetation gradually used deep soil water with a decrease in
precipitation.

The $\delta^{18}O$ value of soil water decreased with the increase of soil depth in

mountainous and oasis areas of arid regions, and the maximum value appeared in the
soil surface layer (Fig. 4). In the three regions, the $\delta^{18}O$ of soil water in the
mountainous area varied greatly from 0 to 30cm (in the HLZ and XYWG) and from 0
to 40cm (in HJX), with the variation range from -4.62 to -7.05 in the HLZ and
XYWG, and from -7.04 to -9.69 in the HJX. Soil $\delta^{18}O$ of the two sampling sites in the
oasis area also varied greatly from 0 to 30cm (WWPD) and 0 to 40cm (HYSSK), with





a variation range of -4.98 to -8.79 in WWPD and -1.80 to -5.88 in HYSSK. In the
desert area (QTH), the $\delta^{18}$O of soil water in the 0-20cm soil layer changed greatly,
ranging from -3.07 to -2.39, indicating that the soil layer in other stations except DTX
had undergone drastic evaporation.

The $\delta^{18}$O values of groundwater in the oasis area (WWPD and DTX) were

similar to the $\delta^{18}$O values of soil water in 50cm and 90cm, respectively, indicating that
groundwater could replenish soil water in these two locations, while the $\delta^{18}$O values
of soil water in other locations were significantly different from those of groundwater,
indicating that groundwater did not replenish soil water in these locations. In addition,
the $\delta^{18}$O values of xylem water at seven sampling sites were close to the $\delta^{18}$O values
of soil water at different depths. the mountain area is 10~20cm (HLZ, XYWG) and
30~40cm (XJX); The oasis area is 10-20cm (WWPD)) and 30-40cm (HYSSK),
respectively. The desert area is 40~50cm (QTH); These results indicate that soil water
is a potential source of water for vegetation at these sites, and that vegetation
gradually uses deep soil water as precipitation decreases. In addition, the $\delta^{18}$O value
of xylem water of maize in the oasis area (DTX) was significantly different from that
of soil water, which may be because irrigation water was the main water source for
farmland in the oasis area (DTX).





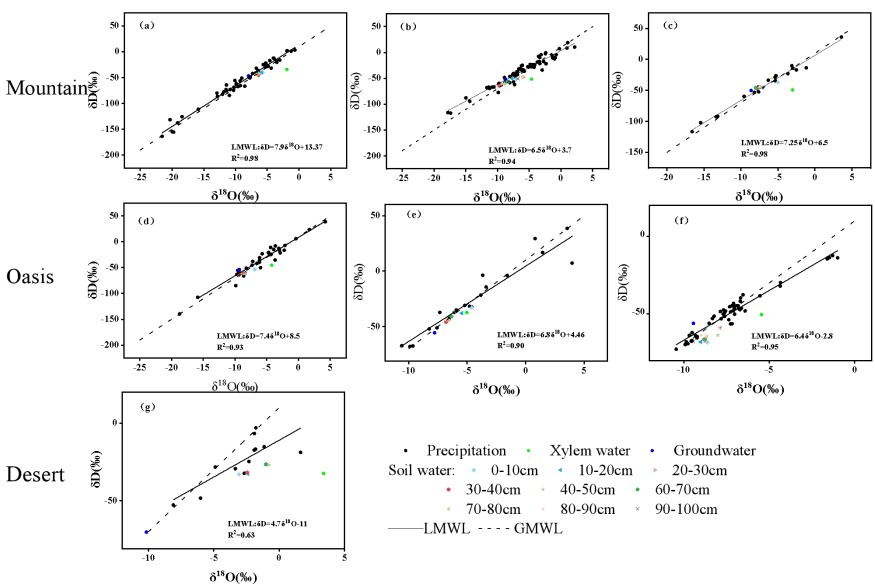

**Fig. 3.** Stable isotopes (δ18O(‰), δD(‰)) of soil water, plant xylem water and groundwater at different depths in mountainous, oasis and desert areas of arid regions. LMWL represents the local atmospheric water line (solid line), and GMWL represents the global atmospheric water line (δH=8δ18O+10). (a)~(f)respectively represent the Hulinzhan, Huajianxiang, Xiyingwugou, Wuweipendi, Hongyashanshuiku, Datanxiang and Qingtuhu.





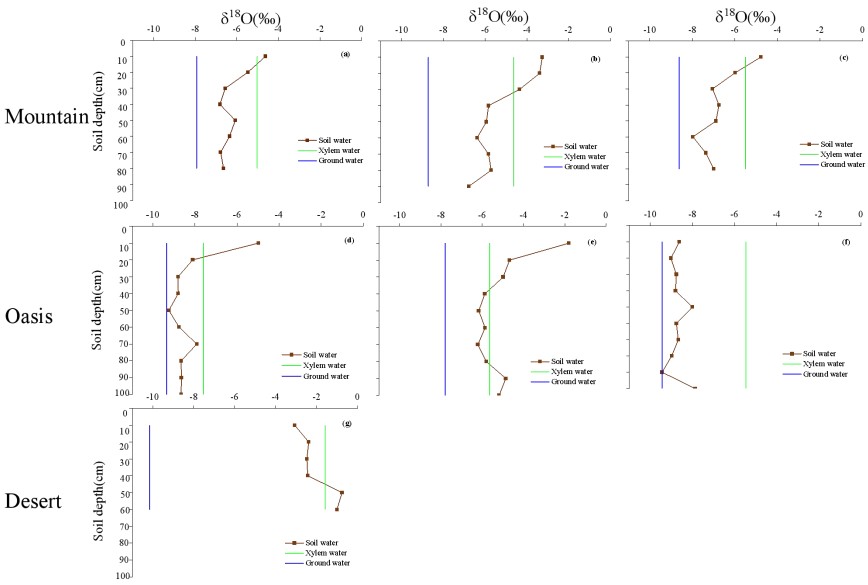

**Fig. 4.** δ¹⁸O values of xylem water, soil water and groundwater in different soil layers of mountainous, oasis and desert areas in the arid region. (a) ~ (f) respectively represent the Hulinzhan, Huajianxiang, Xiyingwugou, Wuweipendi, Hongyashanshuiku, Datanxiang and Qingtuhu.

**3.2 Calculation of vegetation water sources**

The relative contributions of potential water sources to vegetation in seven sites of the mountainous area, oasis and deserts were calculated (FIG. 5). In the mountaious area(HLZ, HJX and XYWG), the vegetation utilization rate of precipitation is 23.1%, 12% and 16.8% respectively, while the utilization rate of soil water is 65.5%, 74.5% and 65% respectively. Because precipitation is the main source of soil water in mountainous areas, 85.6% of vegetation water comes directly or indirectly from precipitation.

Vegetation in the oasis area (WWPD, HYSSK and DTX) uses soil water at 65%,



65.8% and 45.8%, respectively, and groundwater at 18%, 17.7% and 18.1%,
respectively. Surface and underground irrigation water are the main sources of water
for crops in the oasis area (DTX). The reason for the low utilization rate of soil water
by the vegetation in DTX is that the vegetation in this area directly uses river water
and groundwater at a ratio of 37.6%. Therefore, irrigation water directly or indirectly
contributes 83.4% of the water at this sampling point.

Vegetation in the desert area (QTH) uses soil water at a rate of 46.1%, and

directly uses lake water and groundwater at a rate of 37.7%. Around the QTH,
vegetation was formed in the affected area of artificial ecological water transport, and
the ecological water transport directly or indirectly contributed 83.8% of the water
content of the plants.

With the decrease in precipitation, the highest soil water use efficiency of

vegetation in seven sites in mountainous, oasis and desert areas of arid region
gradually shifted from shallow soil layer to deep soil layer. Vegetation in a
mountainous area (HLZ, HJX, XYWG) mainly uses shallow soil water, while
vegetation in an oasis area(HYSSK, DTX) and desert area (QTH) mainly uses middle
and deep soil water.



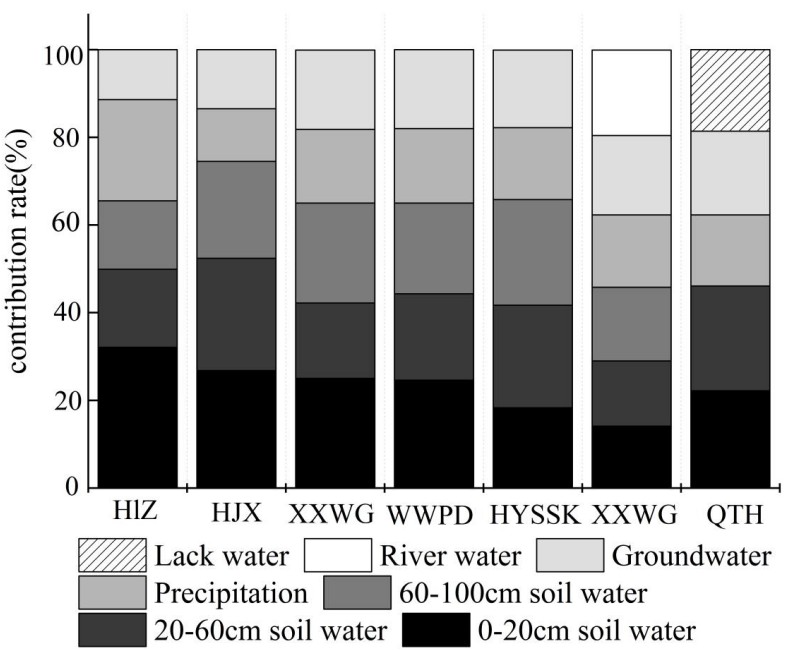

**Figure. 5.** The relative contribution of different potential water sources (soil water, precipitation, groundwater, river water, lake water) to the vegetation of mountainous, oasis and desert areas in arid region. HLZ, HJX, XYWG, WWPD, HYSSK, and DTX represent the Hulinzhan, Huajianxiang, Xiyingwugou, Wuweipendi, Hongyashanshuiku, Datanxiang and Qingt uhu.

**4 Discussion**

**4.1 Effects of precipitation on plant water use strategies**

As one of the main sources of water (Zhao et al., 2018), precipitation is the main factor limiting the growth and development of vegetation in arid and semi-arid areas (Jiang et al., 2017). Moreover, effective precipitation and physiological characteristics of vegetation affect the rate of precipitation utilization by vegetation (Poorter et al., 2019; Sankaran and Staver et al., 2019). Under different precipitation conditions, the


main water sources of vegetation are different. When precipitation is high, the surface
soil water replenished by rain increases, and plants increase their use of shallow soil
water (Lin et al., 1996; Williams & Ehleringer., 2000; Duan et al., 2008). However,
when precipitation decreases, the soil water content decreases significantly, and
shallow soil water cannot meet the needs of vegetation, so deep soil water is sought to
sustain life activities, thus improving the utilization efficiency of deep soil water by
plants (Groisman et al., 1999). The precipitation isotope values at each sampling point
in this study differed due to the influence of precipitation, evaporation source, and
topography. The isotope values of precipitation generally indicated a tendency of
gradual increase from mountainous areas to oasis areas to desert areas. In the 7
sampling points, precipitation decreases from south to north. Although the study area
is located in an arid area, due to the high altitude in the mountainous area,
precipitation can reach 124-698mm, while the precipitation in the oasis area is
83-124mm, and the precipitation in the downstream desert area is 54-82mm (Ma et al.,
2009), thus forming three precipitation gradients. The results showed that the
utilization rate of Qinghai spruce in the mountain forest station was the highest
(23.1%), and the utilization rate of vegetation in other sampling sites was lower than
that in the forest station and similar, with a ratio of about 16%-17%.

Precipitation is an important factor controlling soil water isotopes (Wang et al.,

2017). In arid and semi-arid areas, plants mainly absorb shallow soil water
supplemented by precipitation or deep soil water supplemented by groundwater
(Dodd et al., 1998; Zeng et al., 2013). Under the conditions of rare precipitation, deep



underground water depth or unstable soil water, herbage plants and deep-rooted plants
can provide water through hydraulic uplift to meet the water demand of the formation
(Tang et al., 2018). When the precipitation recharge of soil water cannot meet the
water demand, Water absorption shifts from shallow soil to deep soil (Yang et al.,
2015), and local water cannot meet the growing demand of vegetation, which will
suffer from water stress (Tang et al., 2018). Deeper soil water is generally more
deficient in heavy isotopes than shallow soil water collected at the same location
(Zhou et al., 2019), partly due to the capillary movement of groundwater containing
light isotopes (Rezzoug et al., 2004). In this study, soil water was the main water
source for vegetation in the arid area. With the increase of soil depth, the variation
range of δ18O value of soil water gradually decreased and tended to be stable. The
vegetation utilization rate of soil water in the study area ranged from 45.8 to 74.5%.
In the whole arid region, the soil water content of the mountainous area, oasis and
desert showed a great difference in space. The soil water content of the three regions
was ranked as follows: mountainous area, oasis area and desert area. Soil moisture
content in the region with the altitude of the down trend is obvious, mainly because of
shiyang river basin upstream of high altitude mountainous area precipitation more, is
the recharge area of water resources of the region, and the oasis and desert rely mainly
on the mountains of ice and snow melt water supply, but the condition of oasis area is
better than in the desert region.
In the three regions, along the precipitation gradient, from the mountain area to
the oasis area and then to the desert area, the use of soil water by vegetation gradually



shifted from the shallow layer to the deep layer. Picea crassifolia in the mountain
(HLZ) used 0~20cm of shallow soil water at a ratio of 32.1%, while willow in HJX in
the mountain area, poplar in XYWG in the mountain area and corn in WWPD in the
oasis area slightly reduced the use rate of shallow soil water compared with the HLZ,
which was 26.8%, 25% 24.6%.In the HYSSK in the oasis area, the utilization rate of
shallow soil water by poplar trees in this location is lower (18.3%), but the utilization
rate of medium and deep soil water is increased, with values of 23.4% and 24.1%,
respectively. In DTX of the oasis area, under the conditions of less precipitation and
strong evaporation, irrigation water becomes the main water source for local farmland
vegetation. Therefore, the utilization rate of soil water by local vegetation is lower
than that of other places, and the utilization of deep soil water is slightly higher than
that of shallow and middle soil water. In the QTH in the desert area, the utilization
ratio of the vegetation in this area to the middle and shallow soil water is relatively
average.
**4.2 Effects of human activities on plant water use strategies**
Human activities have an important impact on plant water use patterns in arid
areas, and the impacts mainly occur in the middle and lower reaches. With the further
strengthening of human factors on hydrological control, the water use strategies of
vegetation around the water body of Shiyang River Basin are also affected. Through
the calculation of vegetation water sources at various stations and previous studies on
vegetation water use strategies in arid areas, the influence of human activities on
vegetation water use patterns in the middle and lower reaches of arid areas is



discussed.
Reservoirs are a transitional link between terrestrial and aquatic ecosystems, and
their hydrological changes are vulnerable to local human activities (Naiman and
Décamps et al., 1997; Newman et al., 2006; Tonkin et al., 2018), and the seepage
from reservoirs can have an impact on the water use strategies of vegetation around
reservoirs. The impact is mainly that the reservoir recharges the surrounding soil
water through seepage, which affects the water use strategy of vegetation around the
reservoir, and results showed that the construction of reservoirs had an important
impact on the water consumption strategy of riparian trees in the arid region, and the
influence range of reservoirs on vegetation water absorption pattern was within 2Km.
In the study area of Oasis Hongyashan Reservoir, with the increase in distance,
vegetation increased the utilization of soil water and decreased the utilization of
groundwater.In addition, irrigation has a significant impact on the agricultural water
cycle in arid areas with low precipitation and high evaporation, and in areas with
extreme water scarcity, agricultural water resources account for 80% of total water
resources (Zhu et al., 2021). The sampling site in the oasis area, DTX, has low
precipitation, and agricultural irrigation is a key factor in the existence of the oasis.
Due to anthropogenic irrigation, agricultural vegetation such as maize in the area is
used for irrigation, in addition to precipitation and soil water, and river water and
groundwater are used as the main water source for vegetation in the area. The
vegetation absorbs soil water supplemented by past irrigation water in addition to the
direct use of current irrigation water, and the utilization rate of current irrigation water



is larger at 37.6%. In some terminal lake areas of arid regions, artificial ecological
water transfer is carried out to protect the ecological environment and in these areas,
ecological water is an important water source used by plants, and the water use
strategy of desert plants adapts when the hydrological environment such as
precipitation and groundwater changes (Chen et al., 2017). The ecological water
transfer project launched in 2007 has changed the hydrological conditions in the
surrounding areas of Qingtu Lake in the desert region, which resulted in the changes
in vegetation water use strategies in the catchment area of Qingtu Lake. The study
showed that spatially, the water use of white spurge in this area gradually shifted from
topsoil water to deep soil water as the distance between the sample site and the lake
catchment increased(Jiang et al.,2019). Therefore, we conclude that human activities
control the water use pattern of mid- and downstream vegetation in the arid zone to
some extent.
**4.3 Implications of vegetation water use strategies in different geomorphic units**
**for water resources management**

Water resources in arid and semi-arid areas of social and economic development

and ecological protection play an important role, and space-time distribution of the
heterogeneity of water shortages in arid regions of the northwest means that the
fragile ecological system of the region and the interior of the shiyang river basin has a
unique water cycle, and water resources have significant characteristics, main show is
mountain is forming region, Oases and deserts are dissipation zones (Chen et al.,
2016). In order to meet regional water demand, local water resources can be



supplemented by external water sources, such as inter-regional rivers and
long-distance water transmission channels. When regional water resources are greater
than the maximum demand, no additional water supply is needed (Wang et al., 2008).
The water resources management system based on administrative management should
be established to strengthen the management of Shiyang River basin, so as to promote
the orderly development, effective distribution and rational utilization of water
resources in Shiyang River Basin. The isotopic composition reflects how plants
respond to drought and water scarcity: At the ingestion point (root system),
differences in isotope ratios between plant species are clearly caused by
species-dependent strategies of plants to cope with water stress, through different
utilization of suitable water along the soil profile (shallow or deep) (Yakir and
Yechieli., 1995). This is mainly due to the differences in soil water absorption depth
and the time of stomatal opening in the daily cycle (Gat et al., 2007). The results of
MixSIAR model showed that the vegetation of different geomorphic units in Shiyang
River Basin had different potential water sources, and the utilization ratio of main
water sources was different. In the mountainous area, vegetation has higher utilization
of precipitation and surface soil water and less utilization of groundwater, while the
mountainous area has abundant water resources and provides a continuous water
source for the oasis in the basin, so it is necessary to improve the water connotation
function in the mountainous area and strengthen the construction of water connotation
forest, in addition, in order to reduce evaporation, mountain reservoirs can be built to
abandon the plain reservoirs to reduce the evaporation loss in a large area of the plain.





In the oasis area, agriculture irrigation consumes a large number of water resources.
The main use of irrigation water and farmland vegetation water in deep soil layers, so
it can optimize the structure of planting crops, and using advanced water-saving
irrigation technology, combined with the crop growing period, reasonable
arrangement of irrigation depth and quantity so as to improve the efficiency of
management, to meet the appropriate management of water resources, realize the
sustainable development of agriculture. In the desert area, precipitation is scarce and
evaporation is strong. The decline of groundwater level in this area will speed up the
process of ecological degradation and desertification, especially the vegetation growth
in the lake catchment area is affected by the ecological water transfer. In order to
protect the ecological environment, we can continue to do artificial ecological water
transport, rationally plan the spatial distribution of sand-fixing plants, and improve the
vegetation structure in order to preserve the ecological environment and promote
ecological restoration.
**5 Conclusion**
In this study, δD and δ¹⁸O stable isotope methods were used to study the water
use characteristics of vegetation in mountainous, oasis and desert areas of Shiyang
River basin in arid northwest China. Precipitation and soil water are the main sources
of forest trees in mountainous areas, and the proportion of irrigation water
replenishment for woodland and farmland vegetation in the middle and lower reaches
of the oasis region is high. The desert area forms vegetation in the ecological water
transport area, and the vegetation mainly absorbs the groundwater, soil water and lake



water formed by the ecological water transport. On the whole, plant water use patterns
in mountainous areas are basically not affected by human activities, oasis areas are
mainly affected by irrigation activities and leakage of water conservancy facilities,
and the inland river terminal lake areas are mainly affected by ecological water
transport. As precipitation decreased from mountainous areas to desert areas, the
utilization of soil water by vegetation at different sampling sites gradually shifted
from shallow to deep layers. In addition to the important impact of precipitation on
the growth and development of vegetation in arid areas, human activities also
determine the vegetation water use patterns in the middle and lower reaches of arid
areas through irrigation and artificial ecological water transport. Therefore, basin
management should be strengthened in Shiyang River Basin to promote the orderly
development, effective distribution and rational utilization of water resources in the
basin.
**Acknowledgements**
This research was financially supported by the National Natural Science Foundation
of China(41971036, 41867030), the National Natural Science Foundation innovation
research group science foundation of China (41421061).
**Data Availability Statement**
The data that support the findings of this study are available on request from the
corresponding author, stable isotope data are not publicly available due to privacy or
ethical restrictions.
**Conflict of Interest Statement**



The authors declare no conflicts of interest.
**Contributions**
Siyu Lu: Writing-Original draft preparation; Guofeng Zhu: Writing-Reviewing and
Editing; Rui Li: Data curation; Yinying Jiao: Methodology; Gaojia Meng:
Visualization; Dongdong Qiu: Investigation; Yuwei Liu: Supervision; Lei Wang:
Software; Xinrui Lin: Software; Yuanxiao Xu: Validation; Qinqing Wang: Software;
Longhu Chen: Software.

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
