# Peer review of "Human activities determine vegetation water use in the"

_Biogeosciences, 2023_

## Author Comment (AC1)

**Response report**

Thank you very much for your E-mail of June 3, 2023. We appreciate the editor and reviewer′s constructive comments and suggestions for our manuscript entitled "Human activities determine vegetation water use in the middle and lower reaches of arid areas " with the reference bg-2023-1.

According to the reviewer's comments, we have revised our manuscript carefully and the revised portions have been marked in red in the manuscript track changes version. The main corrections and the response to the reviewer′s comments are as follows.

**Responses to the reviewer's comments**

**Response to Reviewer #1**

This manuscript presents the results of a sampling campaign across seven sites with the aim of determining plant water use sources. However, there are a number of problems.

Foremost, the manuscript is vague throughout. For example, it is not apparent what samples and analyses give rise to the results that they present. It is unclear how many soil samples were taken, at what times of the year, and whether the soil and plant isotope values that they present are from single sampling events, are averages of all the sample they collected, or are a subset of the samples, or what. This is clearly important because soil water and plant water stable isotope values are known to fluctuate throughout the year. Therefore, how can simple results of XX% of plant

water be interpreted to be sourced from ZZ soil depth? Is that the yearly average, or something else?

Below, I list specific comments. Unfortunately, I cannot endorse publication of this manuscript.

Response: Thank you very much for reviewing our article, your comments are very important to us. In this study, we mainly used precipitation, soil, vegetation, and groundwater samples collected at seven stations on different precipitation gradients (mountain, oasis, and desert areas) in the Shiyang River basin from 2017 to 2019 plant growing season (April to November), analyzed the relationships among precipitation, soil water, xylem water, and groundwater isotopes at the seven stations, as well as calculated the sources of vegetation moisture at different stations, and combined with previous The reasons for the different sources of vegetation moisture in different regions are discussed in the context of previous studies.For the collection of precipitation samples, we collected precipitation from all sites for precipitation events during the vegetation growing season (April to November) from 2017 to 2019, and for the collection of soil, vegetation, and groundwater samples, we collected them simultaneously once a month. In addition, for some special sites, such as Datan Township and Qingtu Lake, we collect irrigation water and lake water for analysis because the vegetation in Datan Township is subject to artificial irrigation and the vegetation in Qingtu Lake is affected by the lake water of Qingtu Lake. Below is a plot of the distribution of our sample points.

[Figure]

In the computational analysis, the source of the data we analyzed was not a single sample, but the average $\delta^{18}O$ values of xylem water, soil water, and groundwater in different soil layers collected at each sampling site from 2017 to 2019.

In addition,The main corrections and the response to the reviewer's comments are as follows.

L1: The title asserts that the findings of this study are applicable to the middle and lower reaches of arid areas worldwide. This is not the case. I suggest you add Shiyang River Basin to the title, at least.

Response:Thank you for your suggestion. Based on your comments, we have added the Shiyang River Basin to the title, Specifically:"Human activities determine vegetation water use in the middle and lower reaches of the Shiyang River"

L39: What are examples of these strategies?

Response:The following specific strategies are often used by arid zone vegetation to

cope with extreme conditions:

1.  Reducing water evaporation: Many arid zone plants take measures to reduce water evaporation. For example, during phenological periods, certain herbaceous plants reduce the rate of photosynthesis and reduce the need for water evaporation.Many plants in desert areas are covered with a layer of hair, wax or resin on the bark to reduce water evaporation. Bark can also be used to reduce water loss from trees.

Rzepecki, A., Zeng, F., & Thomas, F. M. (2011). Xylem anatomy and hydraulic conductivity of three co-occurring desert phreatophytes. Journal of Arid Environments, 75(4), 338-345.

2.  Water conservation: Many arid zone plants take measures to conserve water, such as developing stomata. Plants require oxygen and emit carbon dioxide during metabolism.Stomata are plant vents through which plants take in air and emit carbon dioxide. In dry zones, plants conserve water by reducing stomatal size or closing stomata.

Chaves, M. M., Pereira, J. S., Maroco, J., Rodrigues, M. L., Ricardo, C. P. P., Osório, M. L., ... & Pinheiro, C. (2002). How plants cope with water stress in the field? Photosynthesis and growth. Annals of botany, 89(7), 907-916.

3.Storage of water: Some plants are able to store water in their roots, stems, leaves or fruits like a sponge for storing water. For example, cactus has a particularly well-developed water storage mechanism that can store water for several years.

Ogburn, R. M., & Edwards, E. J. (2012). Quantifying succulence: a rapid, physiologically meaningful metric of plant water storage. Plant, Cell & Environment,

35(9), 1533-1542.

4.Finding water: Some plants survive in arid zones by rooting deep underground in search of additional water sources. For example, in the Australian arid grasslands, some plants root underground to a depth of 25 feet and are able to rely on deeply buried sources of groundwater. In addition, some species have dimorphic root systems that allow them to access different water sources in space and time.

Barbeta, A., Mejía‐Chang, M., Ogaya, R., Voltas, J., Dawson, T. E., & Peñuelas, J. (2015). The combined effects of a long‐term experimental drought and an extreme drought on the use of plant‐water sources in a Mediterranean forest. Global change biology, 21(3), 1213-1225.

L54-56: You have not discussed arid lands much in the introduction, but you assert their significance here. I suggest a bit more discussion of arid lands and hydrological issues in them, other than water limitation, especially in regards to vegetation.

Response:Thanks for your suggestions, we have added a discussion of hydrological issues in arid regions to the revised draft, and have reorganized what we previously wrote.Specifically:"Water availability is one of the most important factors for the growth of individual plants in terrestrial ecosystems (Boyer et al.,1982), and plant

survival and activity as well as ecosystem stability are closely related to water availability (Zhou et al.,2019). Precipitation is one of the main sources of water (Zhao et al.,2018), an important climatic factor for vegetation change (Roca et al.,2004), and controls plant community structure, composition and vegetation type (Weltzin et al.,2003). The uneven distribution of precipitation leads to extreme spatial and temporal variability in soil moisture (Antunes et al., 2018), vegetation water use strategies can vary under different precipitation conditions (Miller et al., 2001), and precipitation distribution and water table depth control the effective spatial pattern of soil moisture, which plays a crucial role in plant adaptation and vegetation composition (Zhou et al.,2019).

Arid regions cover 33% of the world's total land area and are characterized by fragile ecosystems and are vulnerable to various natural and human-induced disturbances (Arheimer et al.,2017; Alam et al.,2019). In arid areas, water is the main limiting factor for vegetation development (Porporato et al.,2004) and plant growth is limited due to water scarcity, however plants have strategies to prevent water loss and resist drought (Gupta et al.,2020). Humans have been influencing the hydrological cycle since the beginning of civilization (Zhao et al.,2020), and human activities are a key factor affecting vegetation growth, which is manifested by influencing vegetation types, vegetation degradation, etc. (Klein.,2012; Jiang et al.,2017). In recent years, human activities have changed and continue to change the global and regional environment (Yang et al.,2011), and under the influence of human activities, the hydrological and ecological processes of rivers in arid regions, especially in the

middle and lower reaches, have changed, leading to many ecological and environmental problems (Gibson et al.,2016; Shah et al.,2021). Therefore, it is important to study the effects of human activities on vegetation water use patterns under natural precipitation gradients in inland river basins in arid zones for ecological restoration and water resources management in arid zones."

L61: L61: The lack of isotopic fractionation during plant water transpiration is debateable. Recent research suggests otherwise:

Poca, M., Coomans, O., Urcelay, C., Zeballos, S. R., Bodé, S., & Boeckx, P. (2019). Isotope fractionation during root water uptake by Acacia caven is enhanced by arbuscular mycorrhizas. Plant and Soil, 441, 485-497.

Jiang, H., Gu, H., Chen, H., Sun, H., Zhang, X., & Liu, X. (2022). Comparative cryogenic extraction rehydration experiments reveal isotope fractionation during root water uptake in Gramineae. New Phytologist, 236(4), 1267-1280.

Response:We strongly agree with your suggestion that fractionation during root water uptake should not be ignored. The assumption that the isotopic composition of water in the root zone exactly matches the isotopic composition of plant xylem water was based on early studies that showed no evidence of isotopic fractionation during uptake (Washburn & Smith, 1934) and was supported by several subsequent studies (Zimmermann et al., 1967; Dawson & Ehleringer, 1991; Walker & Richardson, 1991 ).Many early studies only used one isotope (2H or 18O) because their analysis was not as simple, reliable, or conventional as it is today. Except for a small sample

size, this may be the reason why no isotopic fractionation effect was detected. There is evidence that (hydrogen) isotope fractionation during root water uptake may occur in some special plant communities living in saline water or Alkali soil (Ellsworth&Williams, 2007; Lin&Sternberg, 1993).Recent experiments have also shown that isotopic values in xylem water are consistently lower than those in root zone water (Vargas et al. (2017), however, the processes that lead to these isotopic shifts are not clear, although existing studies have suggested some possible explanations. Furthermore, even if fractionation is present during water uptake by vegetation roots, the use of isotopic methods to track water in the soil-plant-atmosphere continuum is not useless.

Washburn, E. W., & Smith, E. R. (1934). The isotopic fractionation of water by physiological processes. Science, 79(2043), 188-189.

Zimmermann, U., Ehhalt, D., & Münnich, K. O. (1968). SOIL--WATER MOVEMENT AND EVAPOTRANSPIRATION: CHANGES IN THE ISOTOPIC COMPOSITION OF THE WATER. Univ., Heidelberg.

Dawson, T. E., & Ehleringer, J. R. (1991). Streamside trees that do not use stream water. Nature, 350(6316), 335-337.

Walker, C. D., & Richardson, S. B. (1991). The use of stable isotopes of water in characterising the source of water in vegetation. Chemical Geology: Isotope Geoscience Section, 94(2), 145-158.

Ellsworth, P. Z., & Williams, D. G. (2007). Hydrogen isotope fractionation during water uptake by woody xerophytes. Plant and Soil, 291, 93-107.

Lin, G., & da SL Sternberg, L. (1993). Hydrogen isotopic fractionation by plant roots during water uptake in coastal wetland plants. In Stable isotopes and plant carbon-water relations (pp. 497-510). Academic Press.

Vargas, A. I., Schaffer, B., Yuhong, L., & Sternberg, L. D. S. L. (2017). Testing plant use of mobile vs immobile soil water sources using stable isotope experiments. New Phytologist, 215(2), 582-594.

L63: This 1966 reference is very old. I suggest looking into the more recent literature. Start with the papers I referenced above.

Response:Thank you for your suggestion, we have updated the latest literature in the revised manuscript.Specifically:"Usually the stable isotope fractionation during plant water uptake is weak to negligible. Thus, xylem water can reflect the isotopic composition of water sources used by most plant species (Wershaw et al., 1966;Dawson & Ehleringer, 1991; Pan et al.,2020).

Dawson, T. E., & Ehleringer, J. R. (1991). Streamside trees that do not use stream water. Nature, 350(6316), 335-337.

Pan, Y. X., Wang, X. P., Ma, X. Z., Zhang, Y. F., & Hu, R. (2020). The stable isotopic composition variation characteristics of desert plants and water sources in an artificial revegetation ecosystem in Northwest China. Catena, 189, 104499."

L69: As well as precipitation frequency and seasonal distribution.

Response:We have added your suggestions to the revised version.Specifically:"Under

water stress conditions, A steady, long-term source of water is essential for plant survival. In addition, the utilization of rainwater by desert plants in arid and semi-arid ecosystems is related to precipitation intensity,precipitation frequency and seasonal distribution."

L70: This is an oddly specific statement about plant species. Are these species relevant to arid areas being discussed here? If so, please explain this to the reader.

Response:Artemisia oleifera and white spurge are mentioned here as the dominant plant species growing in arid desert areas, and the water use by Artemisia oleifera and white spurge under different precipitation conditions was used to respond to the effect of precipitation on vegetation uptake in the arid zone. The dominant vegetation in the desert area in this study is white spurge, and thus we discuss it here.

L86: There is no reference in the text to the map Figure 1. I didn't know there was a map until I got down to the figure.

Response:In response to your comments, we have cited Figure 1 in the text in order to understand the overview of the study area.Specifically:"Shiyang River Basin (36°29′- 39°27′N, 101°41′- 104°16′E) is located in the arid region of northwest China, which is a typical temperate arid inland basin in China. Shiyang River Basin is a temperate continental arid climate, which is controlled by several atmospheric circulations of the westerly belt, eastern monsoon and plateau monsoon (Zhang et al., 2008). The average annual temperature is 8.1℃, and the average annual

precipitation ranges from 82 to 692mm, 80% of which is concentrated in summer. Annual evaporation ranges from 2000 to 2600mm (Wan et al., 2019). Shiyang River is a typical inland water system with a total length of 250 kilometres. From south to north, Shiyang River mainly includes Qilian mountain area in the south, oasis area in the middle and a desert area in the north(Fig. 1). "

L91: This is a huge range in MAP. Could you include some precip contours or something in the map figure to help the reader understand how precip varies across the study area?

Response:Thank you for your suggestion, which was so useful to us that we added a description of the precipitation information for the site to the study area profile.Specifically:"From south to north, Shiyang River mainly includes Qilian mountain area in the south, oasis area in the middle and a desert area in the north(Fig. 1). In addition, the annual precipitation in the three regions is 124-698mm, 83-124mm and 54-82mm from south to north, respectively (Ma et al., 2009), so the soil and vegetation have obvious zonal characteristics (Wang et al., 2012). "

[Figure]

Fig. 1. Overview of the study area and location of sampling points. The names of the sampling points were replaced with their respective abbreviations, from south to north, they were Hulin Station (HLZ), Huajian Township (HJX), Xiying Wugou (XYWG), Wuwei Basin (WWPD), Hongyashan Reservoir (HYSSK), Datan Township (DTX), and Qingtu Lake (QTH).

L93: What is a typical inland water system? What makes Shiyang typical in this sense?

Response:Inland water systems intersect on land to form a relatively independent water cycle system that cannot deliver water directly to the ocean or other water bodies. Inland water systems are characterized by their lack of inflow to the ocean, the diversity of water sources, topographic complexity, irrigated agriculture, the importance of social development, and ecological concerns that need to be addressed. The Shiyang River is known as a typical inland water system because

(1) Source: Shiyang River originates in the eastern part of the Qilian Mountains, north of Lenglongling, Daxue Mountain, the river is 250 kilometers long, the whole

water system from east to west, the main tributaries are Dajing River, Gulang River, Huangyang River, Jinta River, Xiying River, Dongda River and Xida River, etc.. The river system is mainly recharged by rainwater, and also has ice and snow melt water components, with an annual runoff of 1.591 billion cubic meters. The upstream Qilian Mountain area is rich in precipitation, with 64.8 square kilometers of glaciers and residual forest, which is the water recharge of the river.

(2) Flow direction: The overall flow direction of Shiyang River is southwest-northeast.

(3) Topographic features: Shiyang River basin topography is high in the south and low in the north, sloping from west to northeast. The whole watershed can be divided into three major geomorphic units: the southern Qilian Mountains, the southern Qilian Mountains, and the northern desert area.

(4) Irrigated agriculture: The middle reaches of the Shiyang River flow through the corridor plain, forming oases such as Wuwei and Yongchang. The rivers and lakes of the inland water system are often used to irrigate farmland and irrigated agriculture is well developed, so the water resources of the inland water system are extremely important in terms of suitable land use and ecological sustainability.

(5) Ecological and environmental problems: Shiyang River basin is the most densely populated area in northwest China, with the highest degree of water resources development and utilization, the most prominent water use conflicts, and also one of the most serious ecological and environmental problems

Based on the above characteristics, Shiyang River is known as a typical inland water

system.

L98: OK, this precip info is useful. I suggest bringing the earlier statement closer to this info so the reader is not left wondering about it.

Response:Based on your suggestion, we have linked this information to the previous statements in the revised manuscript for better understanding by the reader.Specifically:"From south to north, Shiyang River mainly includes Qilian mountain area in the south, oasis area in the middle and a desert area in the north(Fig. 1). In addition, the annual precipitation in the three regions is 124-698mm, 83-124mm and 54-82mm from south to north, respectively (Ma et al., 2009), so the soil and vegetation have obvious zonal characteristics (Wang et al., 2012)."

L103: OK, I see white thorn mentioned here, perhaps thats why it was mentioned in the intro.

Response:The white thorn mentioned here is the dominant plant species growing in arid desert areas, and its use of water under different precipitation conditions was used to respond to the effect of precipitation on vegetation uptake in arid zones. In this study, white thorn, as the dominant vegetation in the desert area, we mainly analyzed its absorption of water, and thus mentioned it in the introduction.

L106: Fig caption needs to include info about what the map abbreviations represent (study sites).

Response:We have added abbreviated information about the map in the title of Fig 1.Specifically:"Fig. 1. Overview of the study area and location of sampling points. The names of the sampling points were replaced with their respective abbreviations, from south to north, they were Hulin Station (HLZ), Huajian Township (HJX), Xiying Wugou (XYWG), Wuwei Basin (WWPD), Hongyashan Reservoir (HYSSK), Datan Township (DTX), and Qingtu Lake (QTH)."

L148: What is TDF?

Response:We apologize for writing TEF as TDF due to our mistake, and we have corrected it in the revised manuscript. Also, in the MixSIAR model, TEF refers to the Trophic Enrichment Factor (TEF). When analyzing the nutrient sources in stable isotope mixtures, the Trophic Enrichment Factor (TEF) needs to be introduced for correction because the isotope ratios may differ between food sources and the absorption and conversion of stable isotopes by different digestive systems.

In specific applications, TEF values can be obtained from laboratory data or from literature reports. They can be obtained from actual laboratory experimental measurements or from literature reports based on theoretical assumptions and statistical analyses of TEF values. Also, TEF can be obtained on the basis of biological knowledge of the analyzed subject and nutritional ecology, by fixing these values for the whole model as well as for the calculations with great impact.

In the MixSIAR model, TEF is used as one of the input data to calculate the stable isotope ratios of feed sources under different combinations of assumptions. By

introducing the TEF factor, the stable isotope ratios of consumers (e.g., animals) based on absorbed and transformed substances can be calculated more accurately. Therefore, the TEF factor has an important influence on the calculation results of the MixSIAR model.

L154-159: This info is all results material.

Response:In calculating the relative contribution of soil water to vegetation is, three potential water sources were considered, including shallow, middle and deep soil water. For the soil stratification here, reference was made to some other arid zone studies (Zhou et al.,2021), and the delineation results were obtained that the shallow layer (0-20 cm) was influenced by precipitation, irrigation water and evaporation, and the soil water content and isotopic composition of soil water varied greatly; the middle layer (20-60 cm) had relatively little variation in soil water content and isotopic composition of soil water; the deep layer (60-100 cm) had the least variation in soil water content and isotopic composition of soil water. The deep layer (60~100cm) has the least variation in soil water content and the isotopic composition of soil water is more stable.

Zhou Yanqing,Gao Xiaodong,Wang Jiaxin,Zhao Xining. Study on the source of water uptake by the root system of Lycium barbarum in the irrigation area of the Qaidam Basin[J]. Chinese Journal of Ecological Agriculture,2021,29(02):400-409.

L173: Need definition of d-Ecess

Response:Thanks for your suggestion, we have added the definition of d-excess to the methods section.Specifically:"2.4.1 d-excess

d-excess refers to the difference between $\delta$D and $\delta$18O in precipitation due to the different fractionation rates between hydrogen and oxygen isotopes (Dansgaard.,1964). d-excess is related to factors such as water evaporation and precipitation temperature, and is defined as:

d-excess=$\delta$D-8$\delta^{18}$O"

L178: So, the numbers on the X axes are months? i.e. 3 is March?

Response:Yes, in Figure 2, the x-axis is the month and the y-axis is the magnitude of the value. That is, 3 represents the month of March.

L194: This is much too simplistic interpretation of the relationship between precipitation and soil water isotopes. The soil water at the oasis and desert areas is clearly evaporated based on Fig 3. In arid areas, precipitation falling through a dry atmosphere leads to isotopic enrichment (as was discussed, very briefly, earlier). Then what water infiltrates into soil evaporates, other water mixes with the previously existing soil water, etc.

Response:Thanks to your suggestion, we have reinterpreted the relationship between precipitation and soil water isotopes in the revised manuscript.Specifically:"In addition, the isotopic composition of soil water is more enriched than that of precipitation, because the precipitation in this region undergoes evaporation

fractionation in the process of replenishing soil water, which makes the originally small precipitation less replenishing soil water in these two places."

L199: What is xyloxyme ?

Response:We apologize for the spelling of xylem instead of xyloxyme, we have corrected it in the revised manuscript.Specifically:"The $\delta^{18}O$ values of xylem water from Qinghai spruce in HLZ, willow in HJX, white poplar in XYWG, corn in WWPD, poplar in HYSSK, corn in DTX and white prickly tree in QTH were -5.02‰, -4.64‰, -5.51‰, -7.60‰, -5.64‰, -5.45‰, -1.59‰, respectively."

L206: What do you mean by gradually? Along the climate gradient, with soil depth? Thru time?

Response:Gradually here means along the precipitation gradient, i.e. from the mountainous area (124~698mm) to the oasis area (124~698mm) to the desert area (54~83mm), with decreasing precipitation, vegetation gradually utilizes deep soil water. Specifically:vegetation in the mountainous area (Ranger Station, Spartixiang, Xiying Wugou) mainly utilizes shallow soil water, while vegetation in the oasis area (Hongyashan Reservoir, Datan Township) and the desert area (Qingtu Lake) mainly utilizes middle and deep soil water.

L221: Typically, it is the opposite: soil water percolates downward and replenishes (recharges) groundwater. What is the depth to groundwater at the study sites? This is

key information.

Response:Thank for your suggestion, and we fully agree with you about the downward infiltration of soil water to replenish (recharge) groundwater as you mentioned here.In the Shiyang River basin, the depth of groundwater in the Wuwei basin is about 10 m, while the depth of groundwater in Datan Township is about 26 m. In the natural condition, the vegetation does not absorb groundwater, but these two locations are affected by human activities, and groundwater is artificially extracted to irrigate the vegetation in the area, and the groundwater pours into the soil water, so that the soil water is replenished by groundwater. Thus, here is groundwater to soil water replenishment.

L230: Did you sample the irrigation water? What is it H and O isotope values? This is key information.

Response:Yes, among all sampling sites we sampled irrigation water in Datan Township and the sampling dates were 2018: 10 May, 20 June, 10 July and 31 July; 2019: 13 June, 2 July, 15 July, 29 July, 15 August and 31 August. In calculating the vegetation water source, we weighted the D and $^{18}$O isotopic values of the 10 irrigation water events and finally obtained a value of -59.49‰ for $\delta$D and -9.12‰ for $\delta^{18}$O.Sampling information for each irrigation water is as follows:

| Sampling point name | Date | $\delta$D(‰) | $\delta^{18}$O(‰) |
|---|---|---|---|
| Datan Township Irrigation Water | 2018/5/10 | -59.496991 | -8.938996 |
| Datan Township Irrigation Water | 2018/6/20 | -61.496306 | -9.815229 |

| | | | |
|---|---|---|---|
| Datan Township Irrigation Water | 2018/7/10 | -58.816952 | -9.528715 |
| Datan Township Irrigation Water | 2018/7/31 | -57.49039 | -9.891725 |
| Datan Township Irrigation Water | 2019/7/1 | -63.935615 | -9.494959 |
| Datan Township Irrigation Water | 2019/7/15 | -62.585013 | -9.416684 |
| Datan Township Irrigation Water | 2019/6/11 | -54.872018 | -7.682412 |
| Datan Township Irrigation Water | 2019/7/29 | -56.23588 | -8.335484 |
| Datan Township Irrigation Water | 2019/8/15 | -57.289432 | -8.382935 |
| Datan Township Irrigation Water | 2019/8/30 | -62.68606 | -9.581067 |

L234: Figure 3 is hard to read because the symbols are so small. Suggest making the figure bigger with larger symbols.

Response:Thanks for your suggestion, we have enlarged the symbols in Figure 3 in the revised manuscript to make it easier to read.Specifically:"

[Figure]

"

L236: Use meteoric instead of atmospheric.

Response:We apologize for our mistake in writing the local meteoric water line, we have changed it in the revised manuscript based on your comments.Specifically:"LMWL represents the local meteoric water line (solid line), and GMWL represents the global atmospheric water line ($\delta H=8\delta^{18}O+10$)."

L241: What are the sources of the values plotted here? Are these single soil water samples, or are they averages of multiple soil water samples taken at different times?

Response:The source of the data plotted here is not a single sample, but the average $\delta^{18}O$ values of xylem water, soil water, and groundwater for different soil layers collected from 2017 to 2019 at each sampling site.

L251: How did you calculate the 85.6% value?

Response:Thank you for your question in this section, when we analyzed this section, we only considered that in mountainous areas, precipitation is the main source of soil water, and the use of soil water by vegetation in mountainous areas is as high as 65 % to 74.5 %, indicating that the indirect use of precipitation by vegetation is 65 % to 74.5 %, in addition, the direct use of precipitation by vegetation is 12 % to 23.1 %, so we concluded that the direct indirect use by vegetation is 85.6%.However, we did not take into account the recharge of soil water by other water sources, such as meltwater

and groundwater, so the analysis result of 85.6% remains to be verified and this sentence has been deleted in the revised draft.

L258: How do you know that irrigation water contributed 83.4% if you never sampled the irrigation water?

Response:Thank you for your question. For the sampling in Datan Township, we collected precipitation, soil water, xylem water, and irrigation water. The irrigation at this site is artificially extracted from the river and groundwater for irrigation, thus the irrigation water is river water and groundwater.By calculating the relative contribution of potential water sources to the vegetation, it was concluded that the vegetation at the site used 18.1% and 19.5% of soil water and groundwater, thus we concluded that the vegetation at the site used irrigation water directly at 37.6%.In addition, due to the dry climate and low precipitation at the site, large soil water is mainly derived from irrigation water, and the vegetation at the site utilizes 45.8% of soil water, thus we consider that the vegetation at the site indirectly utilizes irrigation water at 45.8%, and thus 83.4% of the water of the vegetation at the site is directly or indirectly derived from irrigation water. However, similar to the previous, we did not consider the contribution of other potential water sources to soil water, and thus the 83.4% analysis result also needs to be verified, and we have removed this sentence in the revised draft.

L265: How are you defining "soil water use efficiency"?

Response:We apologize for the possible lack of clarity here, the soil water utilization rate here refers to the proportion of the soil used by vegetation, we have reworked the sentence in the revised manuscript,Specifically:"With the decrease in precipitation, the highest soil water use of vegetation in seven sites in mountainous, oasis and desert areas of arid region gradually shifted from shallow soil layer to deep soil layer. "

L320-326: This sentence hardly makes sense. This is the first mention of snow or ice.

Response:Thank you for your suggestion, this is indeed the first mention of snow and ice. This statement is mainly discussing the reason for the difference in soil water content between the three regions, as the mountainous areas receive the most precipitation and thus replenish the soil more, while the oasis and desert areas receive less precipitation and replenish the soil less. In addition, oasis and desert areas mainly rely on runoff from mountainous areas for recharge, and we mention snow and ice melt water here mainly because it is the main source of runoff formation in inland river basins, and mountainous areas are runoff formation areas, while oasis and desert are runoff depletion areas.For better understanding, we have modified this sentence in the revised version ,Specifically:"The soil water content in this area has an obvious decreasing trend with the decrease of altitude, mainly because the mountainous area with high altitude in the upper reaches of the Shiyang River Basin has more precipitation and is the water resource recharge area of the whole region, while the oasis and desert area have less precipitation and mainly rely on the water flowing from the mountain area for recharge, and the recharge condition of the oasis area is

better than that of the desert area."

L360: This is the first mention of a reservoir. Did you sample the water in the reservoir?

Response:Thank you for your question, we have previously sampled the water in the Hongyashan Reservoir as well as the surrounding vegetation and soil and analyzed the impact of the presence of the reservoir on the surrounding vegetation. The reason why we put it in the discussion section is because this is the work we did before and it is not shown in the results section, thus we mention it in this part of the discussion.

L364: If the agricultural water sources account for 80% of total water, it seems like it would be important to sample the agricultural water sources (irrigation?) for their isotope values.

Response:Yes, studies have shown that in arid areas with low precipitation and high evaporation, agricultural water resources account for 80% of total water resources. Datan Township mainly extracts river water and groundwater for irrigation of agricultural fields, thus in this study, after each irrigation, we collected the water source (river water or groundwater) for this irrigation for vegetation water source analysis.

L369: So, river water and groundwater are the sources of irrigation water? Did you sample the river water?

Response:Yes, Datan Township mainly extracts river water and groundwater for irrigation of farmland, thus at this point, after each irrigation, we will collect the irrigation water for analysis, for example, when a certain irrigation water is river water, we will sample the river water, and when the irrigation water is groundwater, we will sample the groundwater.

L372: Is this the case for the study area?

Response:Yes, Qingtu Lake is in between Badangilin and Tengri deserts and is a famous coccyx lake. But with the increasing population on both sides of Shiyang River, the lake was swallowed up by the desert on both sides, and after the construction of Hongyashan Reservoir in the 1950s, the water flowing into Qingtu Lake from Shiyang River decreased sharply, and finally dried up completely in 1959.Starting from the 1980s, Minqin County first curbed the desertification through a series of measures such as sand suppression and reforestation, then strengthened the comprehensive management of Shiyang River and water resources deployment. In addition, the ecological water transfer project launched in 2007 has changed the hydrological conditions of the area around Qingtu Lake in the desert area, and after several years of water transfer, Qingtu Lake has been reborn and surface water has re-emerged.

L412: What is connotation?

Response:I'm sorry that what we want to express here is water conservation. You did

not understand it well because of the expression problem. In order to make it easier to understand, we have modified the sentence in the revised manuscript,Specifically:"In the mountainous area, vegetation has higher utilization of precipitation and surface soil water and less utilization of groundwater, while the mountainous area has abundant water resources and provides a continuous water source for the oasis in the basin, so it is necessary to improve the water conservation function in the mountainous area and strengthen the construction of water conservation forest, in addition, in order to reduce evaporation, mountain reservoirs can be built to abandon the plain reservoirs to reduce the evaporation loss in a large area of the plain. In the oasis area, agriculture irrigation consumes a large number of water resources."

We have modified the article systematically and hope to get your valuable comments.

---

## Author Comment (AC2)

**Response report**

Thank you very much for your E-mail of June 3, 2023. We appreciate the editor and reviewer′s constructive comments and suggestions for our manuscript entitled "Human activities determine vegetation water use in the middle and lower reaches of arid areas " with the reference bg-2023-1.

According to the reviewer's comments, we have revised our manuscript carefully and the revised portions have been marked in red in the manuscript track changes version. The main corrections and the response to the reviewer′s comments are as follows.

**Responses to the reviewer's comments**

**Response to Reviewer #2**

This study examined the water use patterns of vegetation along a moderate aridity and elevation gradient in Asia. Well-established methods were used to address patterns of plant water use. The focus is not particularly novel. The title suggests that "human activities" are a major focus of study but really this seems to be a study of vegetation water use patterns at 7 sites. The title suggests that the results can be generalized broadly but this study does provide justification for that.

Conclusions include that "precipitation and soil water" were the main water sources for forest trees, and that irrigation was the main source of water for farmland. These are not particularly compelling results as they are textbook expectations about the hydrologic cycle.

The discussion is lengthy and does not relate well to the data collected in the

study. The paper reads as an observational (rather than process-based) study combined with an extremely broad discussion of human water management. The merging is awkward. While the topic of the paper is appropriate for the journal, this paper primarily re-examines well-established processes without providing new insight. The writing is rough and vague in many places. Given these concerns I cannot recommend publication.

Response:Thank you very much for reviewing our article, your comments are very important for us to improve the quality of the paper. In this study, we mainly used precipitation, soil, vegetation, and groundwater samples collected at seven stations on different precipitation gradients (mountain, oasis, and desert areas) in the Shiyang River basin from 2017 to 2019 plant growing season (April to November), analyzed the relationships among precipitation, soil water, xylem water, and groundwater isotopes at the seven stations, as well as calculated the vegetation moisture sources at different stations, and combined with previous The reasons for the different sources of vegetation moisture in different regions are discussed in the context of previous studies.The innovation of this paper is that instead of studying vegetation water use at a single site, we analyze vegetation water use along the whole Shiyang River basin on different precipitation gradients (mountain (124~698mm), oasis zone (124~698mm), and desert zone (54~83mm)), and discuss the differences in vegetation water sources under different precipitation conditions, so as to further explore the implications for water management strategies in the Shiyang River basin.

Regarding your comment that the discussion is lengthy and not well integrated with

the data collected in the study, our ideas are mainly: in the results section, we show the results analyzed from the data we collected, explore the relationships among precipitation, soil water, xylem water, and groundwater isotopes at the seven sites, and quantify the sources of vegetation water at the seven sites; in the discussion section, we mainly analyze the reasons for the different vegetation water use at different sites, such as precipitation conditions and human activities, in conjunction with previous studies, and finally, thus, further explore the implications for water resources management strategies in the Shiyang River basin.

We have modified the article systematically and hope to get your valuable comments.